# The prevalence of molecular markers of resistance to sulfadoxine-pyrimethamine among pregnant women at first antenatal clinic attendance and delivery in the forest-savannah area of Ghana

**David Kwame Dosoo**[1,2]*, **Jeffrey A. Bailey**[3], **Kwaku Poku Asante**[2], **Felix Boakye Oppong**[2], **Karamoko Niaré**[3], **Jones Opoku-Mensah**[2], **Seth Owusu-Agyei**[1,2,4], **Brian Greenwood**[1], **Daniel Chandramohan**[1]

**1** Department of Disease Control, Faculty of Infectious and Tropical Diseases, London School of Hygiene and Tropical Medicine, London, United Kingdom, **2** Kintampo Health Research Centre, Research and Development Division, Ghana Health Service, Kintampo, Ghana, **3** Department of Pathology and Laboratory Medicine, Warren Alpert Medical School, Brown University, Providence, Rhode Island, United States of America, **4** Institute of Health Research, University of Health and Allied Sciences, Ho, Ghana

\* david.dosoo@kintampo-hrc.org

**Data Availability Statement:** The data underlying the results presented in the study are available

## Abstract

Intermittent preventive treatment during pregnancy with sulfadoxine-pyrimethamine (IPTp-SP) is used to prevent malaria and associated unfavorable maternal and foetal outcomes in pregnancy in moderate to high malaria transmission areas. Effectiveness of IPTp-SP is, however, threatened by mutations in the *Plasmodium falciparum* dihydrofolate reductase *(Pfdhfr)* and dihydropteroate synthase *(Pfdhps)* genes which confer resistance to pyrimethamine and sulfadoxine, respectively. This study determined the prevalence of molecular markers of SP resistance among pregnant women in a high malaria transmission area in the forest-savannah area of Ghana. Genomic DNA was extracted from 286 *P. falciparum*-positive dried blood spots obtained from pregnant women aged ≥18 years (255 at first Antenatal Care (ANC) clinic visit and 31 at delivery from 2017 to 2019) using Chelex 100. Mutations in *Pfdhfr* and *Pfdhps* genes were detected using molecular inversion probes and next generation sequencing. In the *Pfdhfr* gene, single nucleotide polymorphisms (SNPs) were detected in 83.1% (157/189), 92.0% (173/188) and 91.0% (171/188) at codons 51, 59, and 108 respectively in samples collected at first ANC visit, while SNPs were detected in 96.6 (28/29), 96.6% (28/29) and 96.8% (30/31) in isolates collected at delivery. The *Pfdhfr* triple mutant N51I, C59R and S108N (**IRN**) was carried by 80.5% (128/159) and 96.5% (28/29) of the typed isolates collected at ANC visit and at delivery respectively. In the *Pfdhps* gene, SNPs were detected in 0.6% (1/174), 76.2% (138/181), 33.2% (60/181), 1.2% (2/174), 0% (0/183), and 16.6% (27/173) at codons 431, 436, 437, 540, 581 and 613 respectively in samples collected at ANC, and 0% (0/25), 72% (18/25), 40% (10/25), 3.6% (1/25), 0% (0/29) and 7.4% (2/27) in samples collected at delivery. Quadruple mutant *Pfdhfr* N51I, C59R, and S108N + *Pfdhps* A437G (**IRN**-**G**K) was present in 25.8% (33/128) and 34.8% (8/23) of

from URL: https://www.ncbi.nlm.nih.gov/
bioproject/856412 ACCENSION NUMBER:
PRJNA856412.

**Funding:** David Kwame Dosoo received funding
from the Kintampo Health Research Centre,
Research and Development Division, Ghana Health
Service as part of his PhD thesis. Support was also
provided by Prof Jeff A Bailey of Brown University
in whose laboratory training was received by the
Lead Author and sample analysis was carried out.
The funders played a role in the study design, data
collection and analysis, decision to publish and
preparation of the manuscript.

**Competing interests:** The authors have declared
that no competing interests exist.

**Abbreviations:** ANC, Antenatal Care; DBS, Dried
blood spots; IPTp, Intermittent preventive
treatment during pregnancy; IPTp-SP, Intermittent
preventive treatment during pregnancy using
sulfadoxine-pyrimethamine; MIPs, Molecular
inversion probes; *Pfdhfr*, *Plasmodium falciparum
dihydrofolate reductase*; *Pfdhps*, *Plasmodium
falciparum dihydropteroate synthase*; SNPs, Single
nucleotide polymorphisms; SP, Sulfadoxine-
pyrimethamine; WHO, World Health Organisation.

isolates at ANC and at delivery respectively. Quintuple mutant alleles *Pfdhfr* N51I, C59R, and S108N + *Pfdhps* A437G and K540E (**IRN**-**GE**) were detected in 0.8% (1/128) and 4.4% (1/23) of samples collected at ANC and at delivery respectively. No mutations were identified at *Pfdhfr* codons 16 or 164 or *Pfdhps* 581. There is a high prevalence of *Pfdhfr* triple mutant *P. falciparum* infections among pregnant women in the study area. However, prevalence of the combined *Pfdhfr/Pfdhps* quadruple and quintuple mutants **IRN**-**G**K and **IRN**-**GE** respectively prior to commencement of IPTp-SP were low, and no *Pfdhps* A581G mutant was detected, indicating that SP is still likely to be efficacious for IPTp-SP in the forest-savannah area in the middle belt of Ghana.

## Background

An estimated 241 million malaria cases were reported globally in 2020, with 228 million cases (95%) occurring in the World Health Organization (WHO) African Region [1]. Malaria during pregnancy remains a major public health problem, with an estimated 11.6 million out of 33.8 million (34%) pregnancies exposed to malaria infection in areas of moderate to high malaria transmission in Africa, resulting in 819,000 children with low birthweight in 2020 [1].

To minimize the potential unfavorable maternal and foetal outcomes of malaria, such as maternal anaemia, low birthweight, preterm delivery and stillbirth associated with malaria during pregnancy in areas of moderate to high transmission of malaria [2, 3], the WHO recommends the use of intermittent preventive treatment using sulfadoxine-pyramethamine (IPTp-SP) starting early in the second trimester, and given at intervals of at least one month apart until as close as possible to delivery [4]. Efficacy of IPTp-SP is influenced by the level of resistance to SP in the *Plasmodium falciparum* parasite populations [5, 6], and the level of resistance is also influenced by SP use in the population [7, 8]. Increasing resistance of *P. falciparum* to SP is attributable to point mutations in the dihydrofolate reductase *(Pfdhfr)* gene at codons N51I, C59R, S108N and I164L and in the dihydropteroate synthase *(Pfdhps)* gene at codons S436A, A437G, K540E, A581G and 613T/S, which have been reported to confer resistance to pyrimethamine and sulfadoxine respectively [9–11]. Mutations in the *Pfdhfr* and *Pfdhps* genes usually occur in a progressive manner [12]. Accumulation of mutations in codons 51, 59 and 108 of the *Pfdhfr* gene (**IRN** triple mutation) combined with mutations in the *Pfdhps* gene at codon 437 are referred to as a quadruple mutation (**IRN**-**G**K) which confers partial resistance to SP; the triple *Pfdhfr* mutation combined with the double *Pfdhps* mutations A437G and K540E results in a quintuple mutation (**IRN**-**GE**) which confers full resistance; the triple *Pfdhfr* mutation and triple *Pfdhps* mutations (A437G, K540E and A581G) gives a sextuple mutation (**IRN**-S**GEG**A) which confers super resistance [13].

Increasing prevalence of mutations in the *Pfdhfr* and *Pfdhps* genes of *P. falciparum* parasites threatens the use of SP for preventing malaria during pregnancy. WHO recommends that IPTp-SP implementation can be continued in areas of moderate to high malaria transmission when the prevalence of K540E and A581G mutations are <95% and <10%, respectively [14]. Continuous surveillance of SP resistance markers is, therefore, very important to identify any accumulation of mutations in the *Pfdhfr* and *Pfdhps* genes of *P. falciparum* to guide policy on use of SP for IPTp programmes.

In East and Southern Africa, very high levels of quintuple and sextuple mutations have been reported among pregnant women, children and in the general population, almost reaching 100% saturation in some areas and resulting in loss of effectiveness of IPTp-SP [6, 15–17]. In West and Central Africa, however, *Pfdhfr* triple mutation and *Pfdhps* S436A and A437G

mutations are common, but the presence of the K540E and A581G mutations is rare [18–22]. In Ghana, a few studies on SP resistance markers among pregnant women from different locations attending ANC showed **IRN**, **IRN**-S**G**KAA and **IRN**-S**GE**AA mutations covering a range from 71% to 80%, 12–41%, and 0–1% [23–25]. There are data for SP resistance markers in children resident in the forest-savannah zone of Ghana [26, 27] but there are no published data on SP resistance markers in *P. falciparum* isolates obtained from pregnant women in this region.

This study aimed at determining current levels of circulating mutations and haplotypes in the *Pfdhfr* and *Pfdhps* genes among pregnant women prior to commencement of IPTp-SP and at delivery, in an area of high malaria transmission in the forest-savannah transition zone in the middle belt of Ghana.

## Methods

### Study area, participants and sample collection

The study area has been described previously [28]. Briefly, the study was conducted in four adjoining administrative areas, namely the Kintampo North Municipality, Kintampo South District, Nkoranza North District and Nkoranza South Municipality, all within the forest-savannah transitional ecological zone in the middle belt of Ghana. *Anopheles gambiae* and *An. funestus* are the main vectors of malaria transmission in the area. Malaria transmission is high and perennial, with a peak in each year between April and October [27, 29].

This study was part of a trial that evaluated the effectiveness of four or more doses of IPTp-SP in the middle belt of Ghana. Dried blood spot (DBS) samples from a cohort of pregnant women were collected at their first antenatal care (ANC) clinic visit (prior to administration of IPTp-SP) [28] and at delivery [30] to determine the prevalence of molecular markers of resistance to SP.

The study procedures have been described previously [28]. Briefly, informed consent was obtained from pregnant women aged 18 years or above who were visiting the ANC for the first time. A structured questionnaire was completed followed by collection of a venous blood sample (2 mL) into EDTA vacutainer tubes prior to commencement of SP administration. At delivery, 0.2 mL of peripheral and placental blood samples were collected into EDTA microtainer tubes. Thick and thin blood smears were prepared from each sample on the same slide, stained with 10% Giemsa stain after fixing the thin film with absolute methanol and examined by two independent, certified malaria microscopists; discrepancies in presence/absence, species and density were resolved by a third microscopist, according to the method of Swysen et al. [31] and WHO [32]. From EDTA anticoagulated blood samples collected from each participant prior to commencement of IPTp-SP and at their time of delivery, three 50 μL blood spots were preserved on 3MM Whatman filter paper (GE Healthcare, Boston, MA, USA) by air-drying overnight and storing individually with silica gel dessicant in a ziplock bag at room temperature until molecular testing was performed. The DBS from all women who tested positive for *P. falciparum* malaria by microscopy prior to IPTp-SP commencement and/or at delivery were selected for DNA extraction and molecular inversion probe sequencing [33, 34] for markers of *Pfdhfr* and *Pfdhps* gene mutations.

### Molecular analysis

Molecular analysis was performed at the Centre for International Health Research, Brown University, Providence, USA. One 6mm diameter disc of the DBS was obtained using a sterile hole-puncher and placed into a 2.2 mL 96 square well storage plate (Thermo Scientific UK). Genomic DNA extraction was performed using the Chelex method.

*P. falciparum* molecular inversion probe (MIP) design, MIP capture, amplification, sequencing and data processing were performed as previously described [33–35]. Sensitivity and accuracy of the MIP captures were determined using serial dilutions of a control mixture of DNA isolated from the laboratory strains of *P. falciparum* 3D7 (wild-type), HB3, 7G8, and DD2 (mutants) which were mixed at relative frequencies of 67%, 14%, 13%, and 6%, respectively.

Drug resistance markers (amino acid mutations determined from the underlying DNA sequence) examined for this study were *Pfdhfr* A16V, N51I, C59R, S108N/T, I164L for pyrimethamine, and *Pfdhps* A431V, S436A, A437G, K540E, A581G and A613S for sulfadoxine. All genotypes with >1 unique molecular identifiers (UMI) coverage were selected for downstream analysis. Results were classified as wild type or mutant. Samples containing mixed infections (i.e. both wild type and mutant) were considered as mutants. Samples with mutations 51I+59R +108N/T (**IRN**) of the *Pfdhfr* protein and 437G+540E (**GE**) of the *Pfdhps* protein were classified as *Pfdhfr* triple mutants and *Pfdhps* double mutants respectively. Overall classification combined the number of mutations in both *Pfdhfr* and *Pfdhps* genes. Triple *Pfdhfr* mutants + 437G, *Pfdhfr* triple mutant + *Pfdhps* double mutant were classified as quadruple (**IRN**-**G**K) and quintuple (**IRN**-**GE**) mutants respectively.

## Statistical analysis

Data from the sequence analysis were stored in Microsoft Excel. Statistical analysis was performed using Stata 14 (StataCorp, College Station, TX, USA). Data were presented for samples collected at ANC clinic prior to commencement of IPTp-SP or at delivery. Prevalence of SP resistance mutations was defined as the percentage of pregnant women who carried at least one resistant parasite clone [36]. The percentage of mutations in the *Pfdhfr* and *Pfdhps* genes at each time point were calculated as the number of samples with mutation at a specific codon divided by the number of samples successfully genotyped and multiplied by 100, whiles percentage of haplotypes were calculated as the haplotype divided by the number of samples with complete genotype results for the haplotype, multiplied by 100. Differences in carriage of SNPs between samples collected prior to commencement of IPTp-SP and at delivery was assessed using the Chi-square test. A p-value of <0.05 was considered significant.

## Ethical considerations

Ethical approval for the study was obtained from the ethics committees of the Kintampo Health Research Centre (KHRCIEC/2017-9) and the London School of Hygiene and Tropical Medicine (LSHTM Ethics Ref: 12338). Informed consent was obtained from each participant prior to enrolment into the study.

## Results

### Characteristics of study participants

Molecular characterization of resistance markers was undertaken successfully in at least one codon for 255 (80%) of 317 DBS samples from women with malaria parasitaemia at first ANC clinic visit and for 31 (94%) of 33 women with a malaria infection at delivery. Samples for which characterization of resistance markers was unsuccessful was due to inadequate amounts of parasite DNA at time of analysis. There was, however, no difference in characteristics of study participants at enrollment whose resistance markers were successfully characterized and those that were not characterized (S1 Table). About a half of the pregnant women at enrolment or delivery were aged <24 years old. Malaria parasite density range (Geometric Mean Parasite

**Table 1. Socio-demographic characteristics of pregnant women with malaria positive dry blood spots at first antenatal care clinic visit or at delivery that were used in the study.**

| Characteristics | Enrolment, n (%) (N = 255) | Delivery, n (%) (N = 31) |
|---|---|---|
| **Maternal Age (years)** | | |
| ≤24 | 124 (48.6) | 18 (58.1) |
| 25–34 | 98 (38.4) | 11 (35.5) |
| ≥35 | 24 (9.4) | 1 (3.2) |
| Missing | 9 (3.5) | 1 (3.2) |
| **Highest educational level** | | |
| None | 86 (33.7) | 2 (6.5) |
| Primary school | 51 (20.0) | 7 (22.6) |
| Junior High/ Middle School | 74 (29.0) | 13 (41.9) |
| Secondary School or higher | 41 (16.1) | 9 (29.0) |
| Missing | 3 (1.2) | - |
| **Marital status** | | |
| Married/married before | 161 (63.1) | 18 (58.1) |
| Living together with a man/unmarried | 39 (15.3) | 5 (16.1) |
| Single, unmarried | 52 (20.4) | 8 (25.8) |
| Missing | 3 (1.2) | - |
| **Malaria parasite density (parasites/μL of blood)** | | |
| Range | 20–46500 | 82–13246 |
| Geometric mean | 522 | 547 |
| **Temperature >37.5 ˚C** | | |
| Yes | 1 (0.4) | 0 (0.0) |
| No | 241 (94.5) | 31 (100.0) |
| Missing | 13 (5.1) | - |
| Mean number of SP doses taken (SE) | - | 2.9 (0.21) |

SE: Standard Error

Density) at enrolment and delivery were 20–46500 (522) and 82–13246 (547) parasites per microlitre of blood, respectively. Mean number of SP doses taken at time of delivery was 2.9 (SE 0.21) (Table 1).

## Prevalence of mutations in *P. falciparum dhfr* and *dhps* genes

Sequence analysis of the parasite DNA extracted from the blood spots of pregnant women with *P. falciparum* parasitaemia at first ANC clinic visit identified N51I, C59R, and S108N mutations of the *Pfdhfr* gene as 83.1% (157/189), 92.0% (173/188) and 91.0% (171/188) respectively. Among the samples collected at delivery, 96.6% (28/29), 96.6% (28/29) and 96.8% (30/31) carried the mutant N51I, C59R, and S108N alleles respectively. None of the pregnant women harboured parasites with *Pfdhfr* A16V, S108T or I164L mutations at their first ANC clinic visit or at delivery (Table 2).

The mutants I431V, S436A, A437G, K540E and A613S alleles in the *Pfdhps* gene were found in 0.6% (1/181), 76.2% (138/181), 33.2% (60/181), 1.2% (2/174), and 16.6% (29/173) in samples obtained from women with malaria parasitaemia at first antenatal clinic. Among the samples collected at delivery, 72.0% (18/25), 40.0% (10/25), 3.6% (1/28), and 7.4% (2/27) carried the mutant alleles respectively (Table 2). None of the pregnant women harboured parasites carrying the mutant A581G allele at first ANC visit or at delivery.

**Table 2. Prevalence of amino acid mutations translated from nucleotide sequence in the *P. falciparum* *Pfdhfr* and *Pfdhps* genes at first ANC clinic visit and at delivery in the forest-savannah area of Ghana.**

| Mutations | ANC | | | | Delivery | | | | p-value [γ] |
|---|---|---|---|---|---|---|---|---|---|
| | Number genotyped | Number of mutations | Prevalence (%) | 95% CI | Number genotyped | Number of mutations | Prevalence (%) | 95% CI | |
| *Pfdhfr* | | | | | | | | | |
| A16V | 174 | 0 | 0.0 | - | 29 | 0 | 0.0 | - | |
| N51I | 189 | 157 | 83.1 | 77.0–87.8 | 29 | 28 | 96.6 | 77.0–99.6 | 0.059 |
| C59R | 188 | 173 | 92.0 | 87.1–95.2 | 29 | 28 | 96.6 | 77.0–99.6 | 0.385 |
| S108N | 188 | 171 | 91.0 | 85.9–94.3 | 31 | 30 | 96.8 | 78.4–99.6 | 0.275 |
| S108T | 188 | 0 | 0.0 | - | 31 | 0 | 0.0 | - | |
| I164L | 197 | 0 | 0.0 | - | 30 | 0 | 0.0 | - | |
| *Pfdhps* | | | | | | | | | |
| I431V | 181 | 1 | 0.6 | 0.1–3.9 | 25 | 0 | 0.0 | - | 0.709 |
| S436A | 181 | 138 | 76.2 | 69.4–81.9 | 25 | 18 | 72.0 | 50.2–86.8 | 0.643 |
| S436F | 181 | 0 | 0.0 | - | 25 | 0 | 0.0 | - | |
| A437G | 181 | 60 | 33.2 | 26.6–40.4 | 25 | 10 | 40.0 | 22.0–61.2 | 0.498 |
| K540E | 174 | 2 | 1.2 | 0.3–4.5 | 28 | 1 | 3.6 | 4.4–23.7 | 0.325 |
| A581G | 183 | 0 | 0.0 | - | 29 | 0 | 0.0 | - | |
| A613S | 173 | 27 | 16.6 | 10.9–21.9 | 27 | 2 | 7.4 | 1.7–27.2 | 0.260 |

[γ]: $\chi^2$ test for difference between prevalence at first ANC and at delivery

## Prevalence of *Pfdhfr* and *Pfdhps* haplotypes

Prevalence of translated protein haplotypes from *Pfdhfr* codons 51, 59 and 108 and *Pfdhps* codons 436, 437, 540, 581 and 613 is presented in Table 3. Complete *Pfdhfr* 51/59/108 haplotype data was available for 159 and 29 samples collected at first ANC and at delivery respectively. Among samples obtained from pregnant women at first ANC visit prior to commencement of IPTp-SP, 80.5% (128/159), 0.6% (1/159), 9.4% (15/159) and 1.2% (2/159) harboured the triple mutant **IRN**, double mutant **IR**S, N**RN** or **I**C**N** *Pfdhfr* gene haplotype respectively. The wild type haplotype NCS was carried by 7.6% (12/159) of the *P. falciparum* isolates. Among samples collected at delivery, 96.5% (28/29) and 3.5% (1/29) contained *P. falciparum* parasites with the triple mutant **IRN** and the wildtype haplotype NCS respectively.

Complete data for codon 437/540 pairs were available for 146 and 24 samples collected at first ANC and at delivery respectively. The double mutant **GE**, single mutant **G**K and wild-type AK were present at 0.7% (1/146), 32.9% (48/146), 65.7% (96/146) and 4.2% (1/24), 37.5% (9/24), 58.3% (14/24) in samples collected at ANC and at delivery respectively (Table 3).

Complete *Pfdhps* codons 436/437/540/581/613 haplotypes were available for 127 and 22 samples collected at ANC and at delivery respectively (Table 3). Ten (10) and 5 distinct dhps haplotypes were obtained for samples collected at ANC and at delivery respectively. Wildtype haplotype of *Pfdhps*, SAKAA, was present in 2.4% (3/127) of *P. falciparum* isolates at ANC whiles none was detected in isolates at delivery. The double mutant S**GE**AA responsible for sulfadoxine resistance was detected in 0.8% (1/127) of samples collected at first ANC, and 4.5% (1/22) among those collected at delivery (Table 3).

**Table 3. Prevalence of *Pfdhfr* (codons 51/59/108), *Pfdhps* (codons 437/540) and *Pfdhps* (codons 436/437/540/581/613) haplotypes at first ANC and at delivery.**

| Mutations | ANC | | Delivery | | p-value [Y] |
|---|---|---|---|---|---|
| | n/N | Prevalence (%) | n/N | Prevalence (%) | |
| *Pfdhfr* (Codons 51/59/108) | | | | | 0.438 |
| ICN | 2/159 | 1.2 | | | |
| IRN | 128/159 | 80.5 | 28/29 | 96.5 | |
| IRS | 1/159 | 0.6 | | | |
| NCS | 12/159 | 7.6 | 1/29 | 3.5 | |
| NRN | 15/159 | 9.4 | | | |
| NRS | 1/159 | 0.6 | | | |
| *Pfdhps* (codons 436/437/540/581/613) | | | | | 0.900 |
| AAEAS | 1/127 | 0.8 | | | |
| AAKAA | 64/127 | 50.3 | 10/22 | 45.5 | |
| AAKAS | 13/127 | 10.2 | 2/22 | 9.1 | |
| AGKAA | 17/127 | 13.4 | 4/22 | 18.2 | |
| AGKAS | 3/127 | 2.4 | | | |
| SAKAA | 3/127 | 2.4 | | | |
| SAKAS | 1/127 | 0.8 | | | |
| SGEAA | 1/127 | 0.8 | 1/22 | 4.5 | |
| SGKAA | 23/127 | 18.1 | 5/22 | 22.7 | |
| SGKAS | 1/127 | 0.8 | | | |
| *Pfdhps* (Codons 437/540) | | | | | 0.458 |
| AE | 1/146 | 0.7 | | | |
| AK | 96/146 | 65.7 | 14/24 | 58.3 | |
| GE | 1/146 | 0.7 | 1/24 | 4.2 | |
| GK | 48/146 | 32.9 | 9/24 | 37.5 | |

N represents individuals with complete haplotype results

[Y:] $\chi^2$ test for difference between prevalence at first ANC and at delivery

Of the 31 samples collected at delivery, 7 had a positive malaria microscopy result at both first ANC clinic visit and at delivery. The complete genotype for the *Pfdhfr* codons 51, 59 and 108 was obtained for 4 sample pairs (prior to IPTp-SP and at delivery), with all 4 pairs having the triple mutant IRN (100% correlation between the two timepoints) whiles complete genotype for the *Pfdhps* codons 437 and 540 was obtained for 3 sample pairs as AK/GK, AK/AK and GK/GK.

## Prevalence of combined *P. falciparum* dhfr and dhps haplotypes

A total of 128 and 23 samples had a complete set of genotyped results for the *Pfdhfr* codons 51/59/108 and *Pfdhps* codons 437/540 haplotypes at ANC and at delivery respectively. Eleven (11) distinct *Pfdhfr* haplotypes were observed in samples collected at ANC whiles 4 were observed for samples collected at delivery. The IRN-AK haplotype constituted the majority in samples collected at ANC (53.1%, 68/128) and at delivery (56.5%, 13/23), followed by the quadruple mutant IRN-GK which was 25.8% (33/128) at ANC and 34.7% (8/23) at delivery. The quintuple IRN-GE mutation was detected in 0.8% (1/128) and 4.4% (1/23) in samples collected at ANC and at delivery respectively. The wildtype haplotype NCS-AK was present in 5.5% (7/128) at ANC but not was detected at delivery (Fig 1).

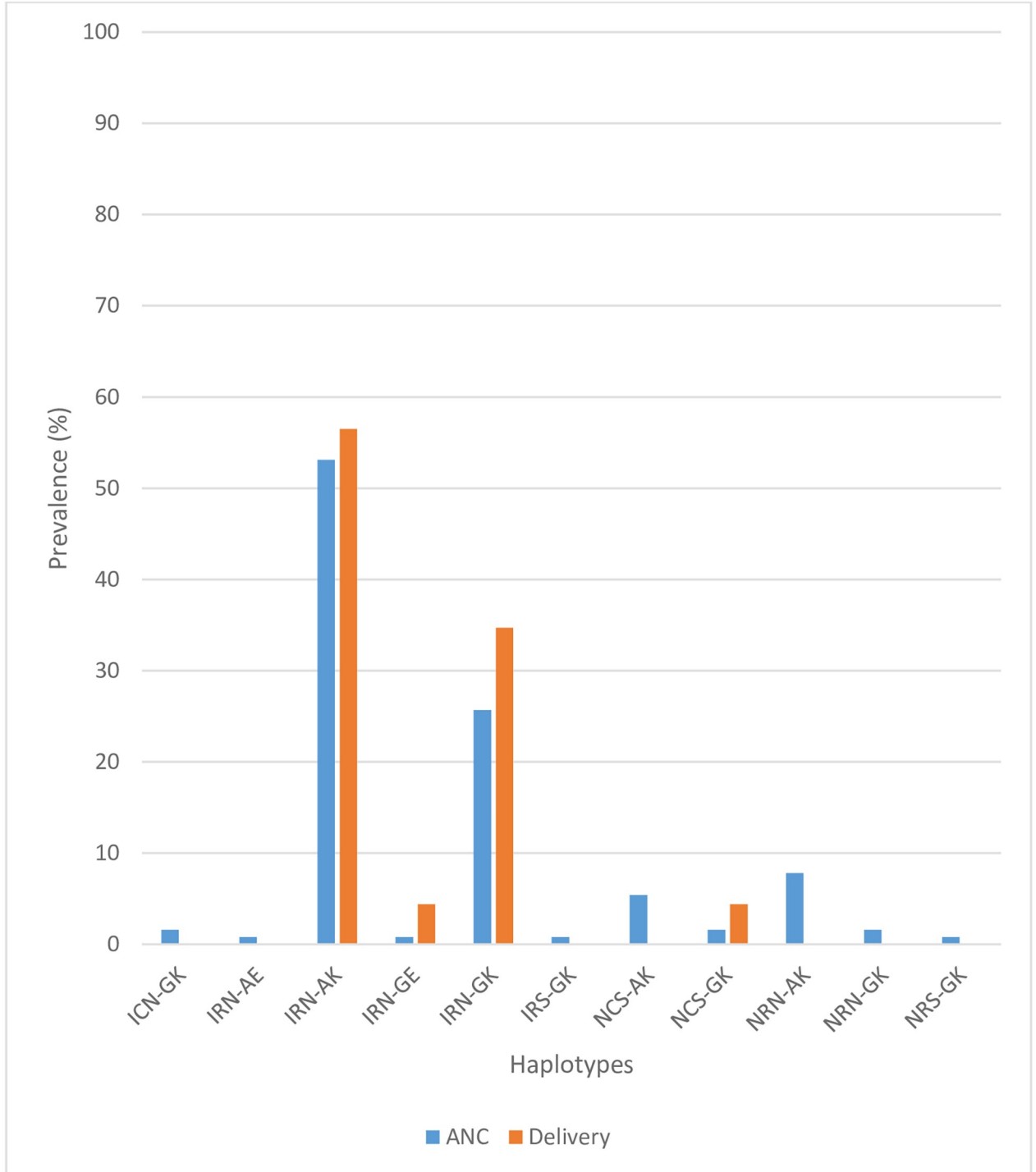

**Fig 1. Prevalence of combined *Pfdhfr* (codons 51/59/108) and *Pfdhps* (codons 437/540) haplotypes at first ANC and at delivery.**

Complete data for the combined *Pfdhfr* 51/59/108 and *Pfdhps* 436/437/540/581/613 was available for 114 and 22 samples collected at ANC and at delivery respectively (Table 4). A total of 21 and 6 distinct haplotypes were obtained from the combined *Pfdhfr* and *Pfdhps* haplotypes. The wild type haplotype (NCS-SAKAA) were carried by 1.7% (2/114) of isolates at

**Table 4. Prevalence of combined *Pfdhfr* (codons 51/59/108) and *Pfdhps* (codons 436/437/540/581/613) haplotypes at first ANC and at delivery.**

| Mutations | Genotype | ANC (N = 114) | | Delivery (N = 22) | |
| --- | --- | --- | --- | --- | --- |
| | | n | Prevalence (%) | n | Prevalence (%) |
| 0 | NCS-SAKAA | 2 | 1.7 | | |
| 1 | NCS-**A**AKAA | 2 | 1.7 | | |
| 1 | NCS-SAKA**S** | 1 | 0.9 | | |
| 1 | NCS-S**G**KAA | 1 | 0.9 | 1 | 4.5 |
| 2 | NCS-**A**AKA**S** | 1 | 0.9 | | |
| 2 | NCS-**AG**KAA | 1 | 0.9 | | |
| 2 | N**R**S-S**G**KAA | 1 | 0.9 | | |
| 3 | I**CN**-S**G**KAA | 2 | 1.7 | | |
| 3 | **IRN**-SAKAA | 1 | 0.9 | | |
| 3 | N**RN**-**A**AKAA | 8 | 7 | | |
| 3 | N**RN**-S**G**KAA | 2 | 1.7 | | |
| 4 | **IRN**-**A**AKAA | 47 | 41.2 | 10 | 45.4 |
| 4 | **IRN**-S**G**KAA | 12 | 10.5 | 4 | 18.2 |
| 4 | N**RN**-**A**AKA**S** | 1 | 0.9 | | |
| 5 | **IRN**-**A**AKA**S** | 11 | 9.7 | 2 | 9.1 |
| 5 | **IRN**-**AG**KAA | 15 | 13.2 | 4 | 18.2 |
| 5 | **IRN**-S**GE**AA | 1 | 0.9 | 1 | 4.6 |
| 5 | **IRN**-S**G**KA**S** | 1 | 0.9 | | |
| 5 | **IR**S-**AG**KAA | 1 | 0.9 | | |
| 6 | **IRN**-**A**AE**A**S | 1 | 0.9 | | |
| 6 | **IRN**-**AG**KA**S** | 2 | 1.7 | | |

first ANC visit and none at delivery. Carriage of the quadruple mutant **IRN**-S**G**KAA was 10.5% (12/114) at first ANC visit and 18.2% (4/22) in the isolates collected at delivery. Triple mutant **IRN** combined with S436A mutation alone represented the highest proportion of mutants (>40%) at both timepoints (Table 3). Quintuple mutations associated with codon 540 (**IRN**-S**GE**AA) were detected in 0.9% (1/112) and 4.6% (1/22) of samples at ANC and at delivery.

## Discussion

Surveillance for *P. falciparum dhfr* and *dhps* resistance markers to sulfadoxine-pyrimethamine among pregnant women is important in evaluating the potential effectiveness of IPTp-SP and in informing policy on malaria control during pregnancy. The recommended approaches for monitoring drug resistance are (i) *in vivo* drug efficacy estimates based on parasite clearance, (ii) *in vitro/ex vivo* drug efficacy assays, and (iii) genotyping of molecular markers. Although determination of molecular markers of resistance is more expensive, time-consuming and laborious compared to *in vitro* and *in vivo* assays for determination of drug resistance, it serves as an excellent complement to the *in vitro* and *in vivo* approaches [37, 38]. This study evaluated carriage of markers of resistance to sulfadoxine and pyrimethamine by describing SNPs in codons 51, 59, 108 and 164 of the *Pfdhfr* gene and in codons 431, 436, 437, 540, 581 and 613 of the *Pfdhps* gene, as well as haplotypes in the *Pfdhfr*, *Pfdhps* and combined *Pfdhfr/Pfdhps* genes among pregnant women prior to commencement of IPTp-SP and at delivery in an area of high malaria transmission in the forest-savannah zone in the middle belt of Ghana to serve as a baseline for monitoring molecular markers of resistance to SP.

It has been proposed that resistance frequencies, as well as prevalence measures, should be evaluated for policy decisions. Whiles prevalence of resistance mutations represent the proportion of infected individuals carrying at least one resistant parasite, frequency of resistance represent the proportion of parasite clones which is carrying a resistant marker [36]. Prevalence of resistance was selected for the analysis of this study as it is a more important way to examine the data for clinical drug effectiveness.

There is no published data on the prevalence of resistance markers to SP among pregnant in the studied area. However, compared to results from samples previously collected from children in the study area, the current study showed a higher prevalence in *Pfdhfr* SNPs at codons 51, 59 and 108 than in samples collected in 2004 (51–66%) [27] but similar to results of samples collected in 2013–2014 (92–95%) [26], whiles the *Pfdhfr* triple mutant **IRN** also showed a marked increase from 31% in 2004 [27] to 81% in the current study. Similar to samples collected in 2013–2014, *Pfdhps* K540E which was not detected in 2004 was present at low levels, but no A581G was detected in the current study.

Prevalence of SNPs at codons 51, 59 and 108 of the *Pfdhfr* gene in *P. falciparum* parasites isolated from pregnant women prior to commencement of IPTp-SP in this study area (83–91%) were similar to the 76–88% prevalence in the Ashanti Region in 2012 [24], the 82–93% in the Greater Accra Region from 2015 and 2017 [25], and the 74–86% in the Western Region [23] all in Ghana. Compared to other African countries, our findings were higher than those reported in 2010 for Bobo-Dioulasso (43–71%) and Nanoro (12–61%) both in Burkina Faso [18, 39], but similar to those reported in Kwale County, Kenya (78–93%) in 2013–2015 [15] and Lagos, Nigeria in 2011 (70–80%) [40] among pregnant women. However, many of these comparator studies were done several years before the current study was undertaken and may have changed subsequently.

Similarly, this study also showed similar prevalence of *Pfdhfr* triple mutants (IRN) in samples collected from first ANC clinic attendants (81%) compared to similar ones in the Ashanti (77%), Western (71%) and Greater Accra (80%) Regions of Ghana between 2010 and 2017 [23–25], Compared to ANC attendees in other countries, our *Pfdhfr* triple mutant results were also similar to those found in Kwale County, Kenya (87%) in 2013–2015 [15], southern Benin (88%) in 2008–2010 [41] and Southeast Nigeria (93%) in 2013–2014 [42]. Our findings were, however, of a higher prevalence than that reported in 2010 from Nanoro in neighbouring Burkina Faso (36%) [18].

The absence of I164L mutations in the *Pfdhfr* gene in our study is consistent with findings of several previous studies involving pregnant women, children or the general population in parts of Ghana [26, 35] and other parts of West Africa [41, 42]. This mutation is generally rare in West and Central Africa, but has been previously reported in malaria cases imported to China from Ghana [43] and also in the Democratic Republic of Congo [44]. The I164L mutation is prevalent in eastern Rwanda and southwestern Uganda [45–47].

Only a small fraction of isolates harboured *P. falciparum* parasites with the A437G mutation in the *Pfdhps* gene in combination with *Pfdhfr* triple mutation, which has been reported to be associated with SP treatment failure. This 26% is much lower compared to other studies in Ghana (92% in 2011–2012) [24] and Nigeria (>90% in 2013–2014) [42]. The low prevalence of the quintuple **IRN**-**GE** mutation in this study (0.8% and 4.4% in samples collected at first ANC visit and at delivery respectively), is supported by other studies in Ghana [23–25] and West Africa [18, 40, 42]. This is an indication that IPTp-SP may still be effective in the study area.

Although not statistically significant, the increase in prevalence of point mutations at codons 51, 59, 108 of the *dhfr* gene and at codons 437 and 540 of the dhps gene in samples collected at delivery compared to those collected prior to commencement of IPTp-SP in this study is consistent with other studies [25, 48]. This increase could be suggestive of increased

selection of SP-resistant *P. falciparum* parasites following SP supplementation during pregnancy [25, 48], or new infections with any of the resistant genotype circulating in the studied area. The average number of SP doses received by the thirty-one women during pregnancy in the study who had malaria parasitaemia at delivery was nearly three doses (slightly below the optimal dose of $\geq$3 doses). Non-compliance with optimal SP dosing is an important contributor to the emergence of drug-resistant *P. falciparum* strains [49]. Increased use of SP has been reported to be associated with clearance of the sensitive *P. falciparum* strains, selection and increased circulation of the resistant phenotype which can be transferred to future generations of the parasite [50, 51].

Detection of the I431V mutation in this study, albeit at a low prevalence, supports a previous finding of the mutation in samples of Ghanaian immigrants in China [43]. Many other previous studies did not report the presence of this mutation in Ghana. This could be due to the other studies not including the I431V in the codons of interest in the *Pfdhps* gene. This mutation was first reported in *P. falciparum* infections imported to the United Kingdom from Nigeria [52], and has subsequently been detected in samples obtained between 2003 and 2015 from pregnant women and children in Nigeria [21] and Cameroon [53]. The impact of I431V mutation on the continuous use of SP for IPTp is, however, yet to be fully assessed [21].

Incremental mutations in the *Pfdhfr* and *Pfdhps* genes are associated with decreased sensitivity of *P. falciparum* to SP. Detection of nearly 10% of A613S mutations which is associated with resistance to SP, in this study for samples collected at first ANC visit and at delivery is similar to findings in the central and eastern regions of Ghana in samples collected from children between 2014 and 2017 [35] and from pregnant women at ANC (2011–2012) in the Ashanti Region of Ghana [24]. These levels of mutations may impact on the use of SP for IPTp. Continuous testing will be necessary to monitor possible distribution and levels of these mutations.

A major strength of this study is that it has described the level of *Pfdhfr* and *Pfdhps* molecular markers of resistance to SP among pregnant women in this area of high malaria transmission which hitherto had not been performed. This study thus provides baseline data for monitoring SP resistance markers in the study area. The study also used the MIPs followed by Next Generation Sequencing which is a highly sensitive and high throughput method for the DNA amplification stage. A limitation, however, is that the number of *P. falciparum* infections at delivery was small and could have resulted in low statistical power to detect an increased prevalence of SP resistance markers following IPTp-SP administration. This reduction in infection numbers itself (i.e. about 20% prevalence of parasitaemia prior to commencement of IPTp-SP [28] against about 3% prevalence at delivery [30] in the study area) suggests the effectiveness of IPTp-SP in clearing *P. falciparum* infections during pregnancy. Also, DNA could not be amplified in a substantial number of DBS, possibly due to DNA degradation. However, the baseline characteristics did not differ between the women's whose samples were successfully sequenced and those that were not.

## Conclusions

There is a high prevalence of *Pfdhfr* triple mutants in the forest-savannah zone in the middle belt of Ghana, but this has not reached saturation levels in pregnant women at their first ANC visit. However, prevalence of the combined *dhfr/dhps* quadruple and quintuple mutants **IRN**-**G**K and **IRN**-**GE** respectively prior to commencement of IPTp-SP were low, and no *Pfdhps* A581G mutations were identified in this study. The findings indicate that SP is still efficacious for use as IPTp in the forest-savannah zone, a high malaria transmission area in the middle belt of Ghana.

## Supporting information

**S1 Table. Socio-demographic characteristics of pregnant women with malaria parasitae-mia at first antenatal care clinic visit whose dried blood spot samples were used in the study and of those women from whom no data for analysis were obtained.**
(DOCX)

## Acknowledgments

We are grateful to the study participants, staff of the Antenatal Care Clinics and Maternity wards at the Kintampo Municipal Hospital and St. Theresa's Hospital where the study participants were recruited. We are also grateful to the heads of the facilities for granting permission for the study to be conducted. We are also grateful to staff of the Seth Owusu-Agyei Medical Laboratory and Jeff Bailey's Laboratory for their support in sample preparation and analysis.

## Author Contributions

**Conceptualization:** David Kwame Dosoo, Kwaku Poku Asante, Seth Owusu-Agyei, Brian Greenwood, Daniel Chandramohan.

**Data curation:** David Kwame Dosoo, Jeffrey A. Bailey, Karamoko Niaré.

**Formal analysis:** David Kwame Dosoo, Jeffrey A. Bailey, Felix Boakye Oppong, Karamoko Niaré.

**Funding acquisition:** Jeffrey A. Bailey, Kwaku Poku Asante, Seth Owusu-Agyei.

**Investigation:** David Kwame Dosoo, Jeffrey A. Bailey, Kwaku Poku Asante, Seth Owusu-Agyei, Brian Greenwood, Daniel Chandramohan.

**Methodology:** David Kwame Dosoo, Jeffrey A. Bailey, Kwaku Poku Asante, Karamoko Niaré, Jones Opoku-Mensah.

**Project administration:** David Kwame Dosoo, Kwaku Poku Asante, Jones Opoku-Mensah, Seth Owusu-Agyei, Brian Greenwood, Daniel Chandramohan.

**Resources:** Jeffrey A. Bailey, Kwaku Poku Asante, Seth Owusu-Agyei, Brian Greenwood, Daniel Chandramohan.

**Supervision:** Kwaku Poku Asante, Seth Owusu-Agyei, Brian Greenwood, Daniel Chandramohan.

**Validation:** David Kwame Dosoo, Jeffrey A. Bailey, Kwaku Poku Asante, Seth Owusu-Agyei, Brian Greenwood, Daniel Chandramohan.

**Visualization:** David Kwame Dosoo, Jeffrey A. Bailey, Kwaku Poku Asante, Seth Owusu-Agyei, Brian Greenwood, Daniel Chandramohan.

**Writing – original draft:** David Kwame Dosoo.

**Writing – review & editing:** David Kwame Dosoo, Jeffrey A. Bailey, Kwaku Poku Asante, Felix Boakye Oppong, Karamoko Niaré, Jones Opoku-Mensah, Seth Owusu-Agyei, Brian Greenwood, Daniel Chandramohan.

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
