## [Decision Letter · Decision Letter 0]

18 Apr 2022

PONE-D-22-03757The prevalence of molecular markers of resistance to sulfadoxine-pyrimethamine among pregnant women at first antenatal clinic attendance and delivery in the forest-savannah area of GhanaPLOS ONE

Dear Dr. DOSOO,

Thank you for submitting your manuscript to PLOS ONE. After careful consideration, we feel that it has merit but does not fully meet PLOS ONE’s publication criteria as it currently stands. Therefore, we invite you to submit a revised version of the manuscript that addresses the points raised during the review process.

We look forward to receiving your revised manuscript.

Kind regards,

Aparup Das, Ph. D.

Academic Editor

PLOS ONE

Journal Requirements:

4. We note that Figure 1 in your submission contain map images which may be copyrighted. All PLOS content is published under the Creative Commons Attribution License (CC BY 4.0), which means that the manuscript, images, and Supporting Information files will be freely available online, and any third party is permitted to access, download, copy, distribute, and use these materials in any way, even commercially, with proper attribution. For these reasons, we cannot publish previously copyrighted maps or satellite images created using proprietary data, such as Google software (Google Maps, Street View, and Earth). For more information, see our copyright guidelines: http://journals.plos.org/plosone/s/licenses-and-copyright.

Reviewers' comments:

Reviewer's Responses to Questions

**Comments to the Author**

1. Is the manuscript technically sound, and do the data support the conclusions?

Reviewer #1: Yes

Reviewer #2: Yes

Reviewer #3: Yes

2. Has the statistical analysis been performed appropriately and rigorously? 

Reviewer #1: Yes

Reviewer #2: Yes

Reviewer #3: Yes

3. Have the authors made all data underlying the findings in their manuscript fully available?

Reviewer #1: Yes

Reviewer #2: Yes

Reviewer #3: Yes

4. Is the manuscript presented in an intelligible fashion and written in standard English?

Reviewer #1: Yes

Reviewer #2: Yes

Reviewer #3: Yes

5. Review Comments to the Author

Reviewer #1: The present study evaluates the prevalence of the marker mutations associated with antimalarial drugs (Sulphadoxine and pyrimethamine) resistance among pregnant Women at ANC and at the time of delivery in Ghana. The study has been designed well and appropriate statistical tools have been used to analyse the data. I congratulate the authors for providing scientific evidence for monitoring the emergence and spread of drug resistance in malaria endemic country of Ghana.

Major comments:

1. How did you calculate the sample size?

2. Further, why there is variation in total number of samples analysed for different mutations at ANC and at the time of delivery?

3. The average doses compliance was 2.9 doses. It is a known fact that poor doses compliance promotes drug resistance. Kindly discuss the situation of drug resistance in light of doses compliance in discussion section.

Minor comments:

1. Page no. 11, line no 207, the value of percentage should be 83% (157/189) in place of “81%.(157/189)”

2. Page no. 11, line no 213, the value of percentage should be 76.2% (138/181) in place of “72.6 % (138.181)”

3. Page no. 11, line no 215, the value of percentage should be 72% (18/25) in place of “76.2 % (18/25)”

Reviewer #2: The authors have analyzed the proportion of mutant alleles of two drug resistant genes namely Pfdhfr and Pfdhps among Plasmodium falciparum infected pregnant women from Ghana. They compared the allele frequencies prior to the commencement of intermitted treatment during pregnancy with sulfadoxine-pyrimethamine and at the time of delivery. Data generated is interesting and important for evaluating the effectiveness of the treatment and to make the informed policies.

However, it would be interesting if authors could mention any correlation between two sets of samples. The sampling details have been given in their two referred publications. However, whether the set of samples screened at the time of first visit and that the time of delivery are same or not is not mentioned. This information could generate a very interesting data like i) what was the status of infection at the time of delivery? Were most of the women uninfected at the time of delivery? If yes, that means the treatment is effective despite the presence of mutant alleles, ii) if the subjects screened at both times are common, what is the scenario of mutant alleles in them at two time points. Such type of comparison could generate more valuable information.

Authors may give more details of the time of sample collection and the set of participants in the methods section.

Reviewer #3: The author investigated the prevalence of SNPs in two groups of pregnant women with Plasmodium falciparum malaria at the time of enrolment and provided with IPTp-SP and at the time of delivery during 2017-19. Such a data for prevalence of SNPs in pfdhfr and pfdhps genes from various malaria endemic areas of Africa including Ghana is available for year 2010-2017. Data for prevalence of SNPs in pfdhfr and pfdhps genes during IPTp-SP scheme is also available from various part of Africa. However, regular surveillance data on prevalence of SNPs in pfdhfr and pfdhps genes is important for the effectiveness of IPTp-SP. It is here recommended the article for publication after some critical addition of facts in it. Critical revisions are mentioned below;

Major revision;

1. The author needs to change the nomenclature of mutants which is correct in Table-2 but written incorrectly everywhere in text and abstract too, e.g., N51I and C59R instead of C51I and N59R respectively in text and abstract.

2. It is not clear and convincing that why there is only two points of collection of DBS, before commencement and at the time of delivery. Is there no case of malaria around second dose or in-between the pregnancy? If no, then why not DBS collected in such point of time and investigated. Such cases where malaria occurred other than these two points of time of collection, should be included in the analysis and discussion.

3. Results of prevalence of SNPs in the group of n=31 where DBS collected at the time of delivery should be discussed with the respective data of prevalence of SNPs in each pregnant women which must have been collected before commencement of IPTp-SP. However, it is not mentioned that DBS for delivery group-n=31 was collected at commencement of IPTp-SP.

4. In discussion section line no-342 mention about increased selection due to IPTp-SP, which seems inappropriate as the prevalence of snps at the time of delivery or any point of time of collection may be a new infection and is random to get infected with any of the resistant genotype circulated in the studied area. It is suggested here that the explanation of any event of mentioned selection should be provided in discussion.

5. Discussion part should highlight the earlier studies providing the prevalence of these snp’s and genotypes in the studied area, to provide insight to the resistant genotypes in circulation.

6. The discussion should provide insight to the fact mentioned in line no-371 that the effectiveness of IPTp-SP is inferred with the smaller number of infections at the climax of pregnancy. This reduction in number should also compared with the in simultaneous number of malaria cases in the studied area in that particular time period to deduce the effectiveness of IPTp-SP. If an area has less resistant pfdhfr-pfdhps genotypes in circulation, that’s mean moderate prevalence of SP-sensitive phenotypes can straight way justify effectiveness of IPTp-SP, like the condition in this study, then why we need evaluation of effectiveness through such rigorous practice. It is understandable that the prevalence of SNPs will certainly affect the IPTp-Sp and how much needed to study the prevalence of SNP’s during IPTp-SP is to be discussed.

Best

6. PLOS authors have the option to publish the peer review history of their article (what does this mean?). If published, this will include your full peer review and any attached files.

Reviewer #1: **Yes: **Dr. Anil Kumar Verma

Reviewer #2: No

Reviewer #3: **Yes: **PRASHANT MALLICK

---

## [Author Response · Author response to Decision Letter 0]

26 May 2022

Dear Editor,

Thank you for considering our paper and for sending it out to reviewers. We have responded below to each of the points relating to the journal requirements and those that the reviewers raised and I hope that you will now consider this paper suitable for publication in the PLoS ONE Journal.

Yours sincerely,

 Journal Requirements:

1 Please ensure that your manuscript meets PLOS ONE's style requirements, including those for file naming. The PLOS ONE style templates can be found at

 and

 This has been done

2 We note that the grant information you provided in the ‘Funding Information’ and ‘Financial Disclosure’ sections do not match.

 Grant number KHRC/ADMIN/2015-20

Awarded to DKD by Kintampo Health Research Centre for his PhD study. The funders had no role in study design, data collection and analysis, decision to publish or preparation of the manuscript.

3 In your Data Availability statement, you have not specified where the minimal data set underlying the results described in your manuscript can be found. PLOS defines a study's minimal data set as the underlying data used to reach the conclusions drawn in the manuscript and any additional data required to replicate the reported study findings in their entirety. All PLOS journals require that the minimal data set be made fully available. For more information about our data policy, please see http://journals.plos.org/plosone/s/data-availability.

 A URL and DOI to the study’s minimal underlying data set is being generated and will be provided following acceptance of the manuscript.

4 . We note that Figure 1 in your submission contain map images which may be copyrighted. All PLOS content is published under the Creative Commons Attribution License (CC BY 4.0), which means that the manuscript, images, and Supporting Information files will be freely available online, and any third party is permitted to access, download, copy, distribute, and use these materials in any way, even commercially, with proper attribution. For these reasons, we cannot publish previously copyrighted maps or satellite images created using proprietary data, such as Google software (Google Maps, Street View, and Earth). For more information, see our copyright guidelines: http://journals.plos.org/plosone/s/licenses-and-copyright.

 This figure has been removed from the submission

Response to the reviewers

Thank you very much for reviewing our manuscript and providing useful comments/suggestions to improve the paper. Please find below responses to your comments. The corrections are in track changes in the revised manuscript. 

 Reviewer #1

 Major comments

1 How did you calculate the sample size?

 A ‘classical’ sample size calculation was not done for this study. Dried blood spots (DBS) from all pregnant women who demonstrated a positive malaria result by microscopy at either time point (prior to commencement of IPTp-SP or at delivery) were included in the molecular testing. The methods section has been updated to read “The DBS from all women who tested positive for P. falciparum malaria by microscopy prior to IPTp-SP commencement and/or at delivery were selected for DNA extraction and molecular inversion probe sequencing for markers of Pfdhfr and Pfdhps gene mutations.” (lines 145-148).

2 Further, why there is variation in total number of samples analysed for different mutations at ANC and at the time of delivery?

 The same number of samples were analysed for the different mutations at each of the time points. However, some of the individual mutations could not be genotyped. The prevalence reported is a percentage of the number of mutations detected out of the number successfully genotyped at each time point. The Methods section has been updated with this description as “The percentage of mutations in the Pfdhfr and Pfdhps genes at each time point were calculated as the number of samples with mutation at a specific codon divided by the number of samples successfully genotyped and multiplied by 100, whiles percentage of haplotypes were calculated as the haplotype divided by the number of samples with complete genotype results for the haplotype, multiplied by 100.” (Lines 179-184)

3 The average doses compliance was 2.9 doses. It is a known fact that poor doses compliance promotes drug resistance. Kindly discuss the situation of drug resistance in light of doses compliance in discussion section

 This has been included in the discussion: “The average number of SP doses received by the thirty-one women during pregnancy in the study who had malaria parasitaemia at delivery was nearly three doses (slightly below the optimal dose of ≥3 doses). Non-compliance with optimal SP dosing is an important contributor to the emergence of drug-resistant P. falciparum strains. Increased use of SP has been reported to be associated with clearance of the sensitive P. falciparum strains, selection and increased circulation of the resistant phenotype which can be transferred to future generations of the parasite.” (Lines 373-379).

 Minor comments

1 Page no. 11, line no 207, the value of percentage should be 83% (157/189) in place of “81%.(157/189)”

 This has been corrected to 83.1% (157/189) in Line 214. 

2 Page no. 11, line no 213, the value of percentage should be 76.2% (138/181) in place of “72.6 % (138.181)”

 This has been corrected to 76.2% (138/181) in Line 220.

3 Page no. 11, line no 215, the value of percentage should be 72% (18/25) in place of “76.2 % (18/25)”

 This has been corrected to 72.0% (18/25) in Line 220.

 Reviewer #2

1 It would be interesting if authors could mention any correlation between two sets of samples.

The sampling details have been given in their two referred publications. However, whether the set of samples screened at the time of first visit and that the time of delivery are same or not is not mentioned. This information could generate a very interesting data like i) what was the status of infection at the time of delivery? Were most of the women uninfected at the time of delivery? If yes, that means the treatment is effective despite the presence of mutant alleles, ii) if the subjects screened at both times are common, what is the scenario of mutant alleles in them at two time points. Such type of comparison could generate more valuable information.

 The same cohort of pregnant women were enrolled prior to commencement of IPTp-SP and followed up to delivery. DBS was prepared for all the women at the 2 time points. The Methods section has been updated (lines 145 – 148) to read, “The DBS from all women who tested positive for P. falciparum malaria by microscopy prior to IPTp-SP commencement and/or at delivery were selected for DNA extraction and molecular inversion probe sequencing for markers of Pfdhfr and Pfdhps gene mutations.” 

Among the successfully genotyped samples, only 7 participants had a DBS sample collected prior to commencement of IPTp-SP and another at delivery, the remaining samples from the two time points were not from the same participants. The results section has been updated with the findings of this analysis as “Of the 31 samples collected at delivery, 7 had a positive malaria microscopy result at both first ANC clinic visit and at delivery. The complete genotype for the Pfdhfr codons 51, 59 and 108 was obtained for 4 sample pairs (prior to IPTp-SP and at delivery), with all 4 pairs having the triple mutant IRN (100% correlation between the two time points) whiles complete genotype for the Pfdhps codons 437 and 540 was obtained for 3 sample pairs as AK/GK, AK/AK and GK/GK). (Lines 260-265). 

Most of the women were uninfected at the time of delivery (i.e. about 20% prevalence of parasitaemia prior to commencement of IPTp-SP against about 3% prevalence at delivery in the study area) Lines 405-407.

2 Authors may give more details of the time of sample collection and the set of participants in the methods section.

 The methods section has been updated to read “From EDTA anticoagulated blood samples collected from each participant prior to commencement of IPTp-SP and at their time of delivery, three 50 µL blood spots were preserved on 3MM Whatman filter paper (GE Healthcare, Boston, MA, USA) by air-drying overnight and storing individually with silica gel dessicant in a ziplock bag at room temperature until molecular testing was performed. The DBS from women who tested positive for P. falciparum malaria by microscopy prior to IPTp-SP commencement and/or at delivery were selected for DNA extraction and molecular inversion probe sequencing for markers of Pfdhfr and Pfdhps gene mutations (Lines 140-148). 

 Reviewer 3

1 The author needs to change the nomenclature of mutants which is correct in Table-2 but written incorrectly everywhere in text and abstract too, e.g., N51I and C59R instead of C51I and N59R respectively in text and abstract.

 These have been changed to N51I and C59R.

2 It is not clear and convincing that why there is only two points of collection of DBS, before commencement and at the time of delivery. Is there no case of malaria around second dose or in-between the pregnancy? If no, then why not DBS collected in such point of time and investigated. Such cases where malaria occurred other than these two points of time of collection, should be included in the analysis and discussion.

 Thank you for your comment. The authors agree that there will be cases of malaria between enrolment (prior to commencement of IPTp-SP) and delivery, and it would have been helpful to determine mutations between the two time points. However, the main objective of this study was to determine prevalence of circulating mutations in the studied area before commencement of IPTp-SP (that can affect the effectiveness of IPTp-SP) and at delivery. Although DBS samples were collected at the ANC visits between commencement of IPTp-SP and delivery, molecular analysis was not carried out on these due to resource constraints. 

3 Results of prevalence of SNPs in the group of n=31 where DBS collected at the time of delivery should be discussed with the respective data of prevalence of SNPs in each pregnant women which must have been collected before commencement of IPTp-SP. However, it is not mentioned that DBS for delivery group-n=31 was collected at commencement of IPTp-SP.

 DBS was collected for all participants before commencement of IPTp-SP and at delivery. However, only the DBS of participants with a positive malaria microscopy result was selected for the molecular analysis, as stated under methods (Lines 145-148): “The DBS from women who tested positive for P. falciparum malaria by microscopy prior to IPTp-SP commencement and/or at delivery were selected for DNA extraction and molecular inversion probe sequencing for markers of Pfdhfr and Pfdhps gene mutations.” It was, therefore, not possible to match results of SNPs in the 31 samples collected at delivery to their samples collected at enrolment, as only 7 of the 31 participants with a positive malaria microscopy result at time of delivery had a positive microscopy result prior to commencement of IPTp-SP. 

Please see the response above to a similar question from Reviewer 2 (Point 1) on similarities between samples collected at commencement of IPTp-SP and at delivery. 

4 In discussion section line no-342 mention about increased selection due to IPTp-SP, which seems inappropriate as the prevalence of snps at the time of delivery or any point of time of collection may be a new infection and is random to get infected with any of the resistant genotype circulated in the studied area. It is suggested here that the explanation of any event of mentioned selection should be provided in discussion.

The discussion has been revised in Lines 370-373 as follows: “This increase could be suggestive of increased selection of SP-resistant P. falciparum parasites following SP supplementation during pregnancy, or new infections with any of the resistant genotypes circulating in the studied area.”

5 Discussion part should highlight the earlier studies providing the prevalence of these snp’s and genotypes in the studied area, to provide insight to the resistant genotypes in circulation.

 There are no published data on SP resistance markers available for pregnant women in the study area. A comparison with SP resistance markers in samples collected from children in earlier studies in the study area has been included in the discussion (Lines 327-334) as follows: “There is no published data on the prevalence of resistance markers to SP among pregnant in the studied area. However, compared to results from samples previously collected from children in the study area, the current study showed a higher prevalence in Pfdhfr SNPs at codons 51, 59 and 108 than in samples collected in 2004 (51 – 66%) but similar to results of samples collected in 2013-2014 (92 – 95%), whiles the Pfdhfr triple mutant IRN also showed a marked increase from 31% in 2004 to 81% in the current study. Similar to samples collected in 2013-2014, Pfdhps K540E which was not detected in 2004 was present at low levels, but no A581G was detected in the current study.”

6 The discussion should provide insight to the fact mentioned in line no-371 that the effectiveness of IPTp-SP is inferred with the smaller number of infections at the climax of pregnancy. This reduction in number should also compared with the in simultaneous number of malaria cases in the studied area in that particular time period to deduce the effectiveness of IPTp-SP. If an area has less resistant pfdhfr-pfdhps genotypes in circulation, that’s mean moderate prevalence of SP-sensitive phenotypes can straight way justify effectiveness of IPTp-SP, like the condition in this study, then why we need evaluation of effectiveness through such rigorous practice. It is understandable that the prevalence of SNPs will certainly affect the IPTp-Sp and how much needed to study the prevalence of SNP’s during IPTp-SP is to be discussed.

 A comparison of the prevalence of malaria infections at delivery has been made with the prevalence of infections at first ANC clinic visit (prior to commencement of IPTp-SP in the discussion section (Lines 405-407). This now reads “This reduction in infection numbers itself (i.e. about 20% prevalence of parasitaemia prior to commencement of IPTp-SP against about 3% prevalence at delivery in the study area) suggests the effectiveness of IPTp-SP in clearing P. falciparum infections during pregnancy.” 

The discussion section has also been updated to address the need for this rigorous work in Lines 308-320 as follows: “The recommended approaches for monitoring drug resistance are (i) in vivo drug efficacy estimates based on parasite clearance, (ii) in vitro/ex vivo drug efficacy assays, and (iii) genotyping of molecular markers. Although determination of molecular markers of resistance is more expensive, time-consuming and laborious compared to in vitro and in vivo assays for determination of drug resistance, it serves as an excellent complement to the in vitro and in vivo approaches. This study evaluated carriage of markers of resistance to sulfadoxine and pyrimethamine by describing SNPs in codons 51, 59, 108 and 164 of the Pfdhfr gene and in codons 431, 436, 437, 540, 581 and 613 of the Pfdhps gene, as well as haplotypes in the Pfdhfr, Pfdhps and combined Pfdhfr/Pfdhps genes among pregnant women prior to commencement of IPTp-SP and at delivery in an area of high malaria transmission in the forest-savannah zone in the middle belt of Ghana to serve as a baseline for monitoring molecular markers of resistance to SP.”

---

## [Editor Report · Decision Letter 1]

5 Jul 2022

The prevalence of molecular markers of resistance to sulfadoxine-pyrimethamine among pregnant women at first antenatal clinic attendance and delivery in the forest-savannah area of Ghana

PONE-D-22-03757R1

Dear Dr. DOSOO,

We’re pleased to inform you that your manuscript has been judged scientifically suitable for publication and will be formally accepted for publication once it meets all outstanding technical requirements.

Kind regards,

Aparup Das, Ph. D.

Academic Editor

PLOS ONE
---

## [Editor Report · Acceptance letter]

29 Jul 2022

PONE-D-22-03757R1 

The prevalence of molecular markers of resistance to sulfadoxine-pyrimethamine among pregnant women at first antenatal clinic attendance and delivery in the forest-savannah area of Ghana 

Dear Dr. Dosoo:

I'm pleased to inform you that your manuscript has been deemed suitable for publication in PLOS ONE. Congratulations! Your manuscript is now with our production department. 

Kind regards, 

on behalf of

Dr. Aparup Das 

Academic Editor

PLOS ONE